# Primary and Emergency Care Use: The Roles of Health Literacy, Patient Activation, and Sleep Quality in a Latent Profile Analysis

**DOI:** 10.3390/bs15060724

**Published:** 2025-05-24

**Authors:** Dietmar Ausserhofer, Verena Barbieri, Stefano Lombardo, Timon Gärtner, Klaus Eisendle, Giuliano Piccoliori, Adolf Engl, Christian J. Wiedermann

**Affiliations:** 1Institute of General Practice and Public Health, Claudiana—College of Health Professions, 39100 Bolzano, Italy; 2Claudiana Research, Claudiana—College of Health Professions, 39100 Bolzano, Italy; 3Provincial Institute for Statistics of the Autonomous Province of Bolzano—South Tyrol (ASTAT), 39100 Bolzano, Italy; 4Directorate, Claudiana—College of Health Professions, 39100 Bolzano, Italy

**Keywords:** primary healthcare utilization, latent profile analysis, health literacy, patient activation, sleep quality, healthcare engagement, South Tyrol

## Abstract

Background/Objectives: Healthcare utilization is a behavioral phenomenon influenced by psychosocial factors. This study took place in South Tyrol, a culturally diverse autonomous province in northern Italy, and aimed to identify latent profiles of primary healthcare users based on health literacy, patient activation, sleep quality, and service use, and to examine the sociodemographic and health-related predictors of profile membership. Methods: A cross-sectional survey was conducted with a representative adult sample (*n* = 2090). The participants completed the questionnaire in German or Italian. Latent profiles were identified via model-based clustering using Gaussian mixture modeling and four z-standardized indicators: total primary healthcare contacts (general practice and emergency room visits), HLS-EU-Q16 (health literacy), PAM-10 (patient activation), and B-PSQI (sleep quality). The optimal cluster solution was selected using the Bayesian Information Criterion (BIC). Kruskal–Wallis and chi-square tests were used for between-cluster comparisons of the data. Multinomial logistic regression was used to examine the predictors of cluster membership. Results: Among the 1645 respondents with complete data, a three-cluster solution showed a good model fit (BIC = 19,518; silhouette = 0.130). The identified profiles included ‘Balanced Self-Regulators’ (72.8%), ‘Struggling Navigators’ (25.8%), and ‘Hyper-Engaged Users’ (1.4%). Sleep quality could be used to differentiate between different levels of service use (*p* < 0.001), while low health literacy and patient activation were key features of the high-utilization groups. Poor sleep and inadequate health literacy were associated with increased healthcare contact. Conclusions: The latent profiling revealed distinct patterns in health care engagement. Behavioral segmentation can inform more tailored and culturally sensitive public health interventions in diverse settings such as South Tyrol.

## 1. Introduction

Healthcare utilization is an intricate phenomenon shaped by medical needs, social determinants, and behavioral factors. Traditional models of healthcare access emphasize demographic and socioeconomic determinants ([2]). Emerging research has highlighted the important role of psychosocial attributes such as health literacy, patient activation, and sleep quality in influencing healthcare engagement patterns ([17]; [31]). These factors not only affect how individuals interact with healthcare services but also contribute to disparities in utilization, ranging from preventive visits to emergency care.

Health literacy, defined as the ability to access, comprehend, and apply health information for appropriate decision making, has been widely recognized as a key determinant of healthcare behavior ([39]). Studies indicate that individuals with limited health literacy tend to engage in more frequent primary and urgent care services visits, possibly because of difficulties in self-managing chronic conditions or understanding medical guidance ([33]; [34]). Conversely, low health literacy is associated with lower uptake of preventive services and increased emergency department (ED) utilization due to avoidable complications ([5]; [18]). The European Health Literacy Survey Questionnaire (HLS-EU-Q) has shown that nearly half of European adults exhibit problematic or inadequate health literacy, underscoring the public health relevance of this issue ([38]).

Closely linked to health literacy is patient activation, which refers to an individual’s confidence, knowledge, and skills in managing their health and navigating the health care system ([20]). The Patient Activation Measure (PAM) has been used to categorize individuals into levels of activation, with lower scores predicting higher healthcare utilization, including frequent general practitioner (GP) visits, ED admissions, and hospitalizations ([8]; [26]). Low activation is associated with poorer adherence to treatment regimens and increased healthcare costs, while highly activated patients are more likely to engage in preventive care and self-management ([12]; [17]). Importantly, research suggests that health literacy mediates the relationship between patient activation and healthcare utilization, highlighting the interplay of these psychosocial factors ([19]).

Sleep quality, as measured using instruments such as the Pittsburgh Sleep Quality Index (PSQI), is another critical factor influencing healthcare utilization. Poor sleep is strongly correlated with chronic disease burden and mental health comorbidities, often leading to higher primary and urgent care services consultations, ED visits, and hospitalizations ([24]; [31]). Insomnia has been associated with an increased risk of hospitalization and healthcare costs ([3]; [41]). Moreover, sleep disturbances can exacerbate difficulties in healthcare engagement by impairing cognitive function, reducing health information processing ability, and increasing vulnerability to stress-related health behaviors ([25]).

Despite the growing recognition of these factors as key drivers of healthcare utilization, most studies have examined their effects independently. However, latent profile analysis (LPA) offers a methodological approach for identifying distinct patient subgroups that exhibit shared characteristics across multiple psychosocial domains. Prior research employing latent class analysis has successfully segmented populations based on health literacy profiles ([1]), patient activation clusters ([43]), and sleep-related phenotypes ([36]), revealing different patterns in healthcare engagement. These findings suggest that an integrated data-driven classification of individuals may provide insights into public health interventions.

South Tyrol represents a unique setting for studying healthcare engagement due to its officially bilingual population (German- and Italian-speaking) and the coexistence of two linguistic–cultural communities within a shared regional healthcare system. As an autonomous province of Italy, South Tyrol holds significant legislative and administrative competence in healthcare organization and public health delivery ([42]). This context enables comparative research under controlled service provision conditions, offering insights into how individual-level behavioral and psychosocial traits intersect with sociolinguistic factors. Additionally, previous population-based studies in South Tyrol have shown significant variability in health literacy and healthcare trust across language groups, underscoring the need for region-specific behavioral health analyses.

The present study applied LPA to classify a representative adult population in South Tyrol, Italy, based on healthcare utilization patterns and key psychosocial factors, namely health literacy, patient activation, and sleep quality. By identifying subgroups with distinct utilization behaviors, we aimed to provide empirical evidence for targeted intervention strategies. Specifically, we seek to determine whether certain profiles of individuals, such as frequent healthcare users with poor sleep quality or low-literacy individuals who avoid care, can be distinguished and addressed through tailored public health policies (Figure 1).

By integrating these dimensions, our study contributes to the growing body of evidence on the behavioral and cognitive determinants of healthcare utilization. Understanding latent healthcare user profiles will enable policymakers and healthcare providers to design personalized interventions that improve patient engagement, optimize healthcare service distribution, and enhance population health outcomes.

## 2. Materials and Methods

### 2.1. Study Design and Setting

This cross-sectional study builds on a previously published survey on health information use and trust in South Tyrol’s adult population ([42]). A stratified random sampling approach was used to recruit participants aged ≥ 18 years from the general population of the region. The study was conducted within the framework of primary and urgent care services research at the Institute of General Practice and Public Health at the Claudiana College of Health Professions. Data collection was performed using a structured, self-administered questionnaire available in German and Italian, as previously described ([42]).

### 2.2. Questionnaire

The survey instrument included items on sociodemographic factors, healthcare access and utilization (e.g., frequency of GP and emergency room visits), and core psychosocial dimensions, such as health literacy (HLS-EU-Q16) ([28]; [32]; [38]), patient activation (PAM-10) ([7]; [16]; [27]), and sleep quality (B-PSQI). The same data source was used in an earlier study that analyzed patterns of health information behavior and trust in the healthcare system ([42]).

The choice of instruments was guided by feasibility, validation status, and relevance to target constructs. The HLS-EU-Q16 was selected for its grounding in the European health literacy model and validated German and Italian versions, enabling comparability across South Tyrol’s linguistic groups. While alternatives like the Newest Vital Sign (NVS) ([40]) and the Single-Item Literacy Screener (SILS) ([29]) exist, they offer narrower constructs and limited multilingual validation. For patient activation, PAM-10 was preferred over PAM-13 due to space constraints and proven reliability in diverse settings, including Italian and German populations ([7]; [16]; [27]). These tools offered a balance between brevity and construct validity, facilitating behavioral profiling in multilingual population surveys.

Sleep quality was assessed independently from an earlier study on health information behavior. For this purpose, a validated short form of the Pittsburgh Sleep Quality Index (PSQI) was used based on the abbreviated version proposed by [35] ([35]). The brief PSQI (B-PSQI) includes six items selected from the original instrument to capture the five core dimensions of sleep experience: subjective sleep quality, sleep latency, sleep duration, habitual sleep efficiency, and sleep disturbances. Each dimension was rated on a 0 to 3 scale, yielding a total score from 0 to 15, with higher values indicating poorer sleep quality. A global score above 5 is commonly used as a cutoff to identify individuals with clinically relevant sleep problems. In this study, the six-item version was implemented in both German and Italian using previously validated German and Italian versions of the PSQI. Internal consistency was acceptable across both language groups, with Cronbach’s alpha coefficients of 0.77 and 0.74, aligning well with published values for the Italian version ([11]) and the German version ([23]), respectively.

### 2.3. Statistical Analysis

Before conducting statistical analyses, the dataset was preprocessed to ensure consistency and accuracy. Missing values in the key variables required for LPA were handled using listwise deletion. Continuous variables were standardized through z-transformation to allow comparability across different scales. Missing data were handled using listwise deletion. Implausible values of the primary and emergency care services utilization variables (GP and ER visits) were assessed. To address extreme values, the 99th percentile threshold was applied and values exceeding this threshold were replaced with the 99th percentile value to correct for outliers. No extreme values requiring correction were found for the patient activation, health literacy, or sleep quality variables. After these preprocessing steps, the final dataset was used to identify the latent classes.

#### 2.3.1. Latent Profile Analysis

Model-based clustering analysis was conducted using Jeffreys’ Amazing Statistics Program (JASP; University of Amsterdam, Amsterdam, The Netherlands) to identify latent subgroups within the adult population based on self-reported health-related behavioral and cognitive variables. The analysis included four continuous indicators: number of total healthcare contacts, health literacy score, patient activation score, and sleep quality score.

The model-based clustering approach in JASP fits a series of Gaussian mixture models (GMMs) using expectation-maximization (EM) algorithms, selecting the best-fitting model based on the Bayesian Information Criterion (BIC). The assumption of ellipsoidal clusters with equal volumes and shapes was used. Solutions with two–six clusters were then tested. The optimal number of clusters was determined based on the lowest BIC value and interpretability of the resulting classes. Cluster membership was saved as a categorical variable for the downstream comparative analysis.

#### 2.3.2. Descriptive Statistics and Comparisons

To characterize the identified clusters, descriptive statistics were computed for the total sample and for each class separately. Means, standard deviations, medians, interquartile ranges, and 95% confidence intervals were calculated for continuous variables, whereas categorical variables were summarized using frequency counts and percentages. Group differences across the latent classes were assessed using appropriate statistical tests. Because the Shapiro–Wilk test indicated that most variables were not normally distributed, non-parametric methods were applied. The Kruskal–Wallis test was used as an alternative to ANOVA to compare continuous variables between clusters. Where significant differences were found, Mann–Whitney U tests were applied for post hoc pairwise comparisons without correction. Categorical variables were compared using Pearson’s chi-square test, and Fisher’s exact test was used in cases in which the expected cell counts were low.

#### 2.3.3. Multinomial Logistic Regression Analysis

To examine the associations between sociodemographic and health-related characteristics and behavioral cluster membership, a stepwise multinomial logistic regression was performed using likelihood ratio (LR) criteria to select predictors of cluster membership. Variables were entered and removed based on the significance thresholds for entry (*p* < 0.05) and removal (*p* > 0.10). The procedure uses a single-step rule and is constrained to a maximum of 100 iterations. Cluster 1 (‘Balanced Self-Regulators’) served as the reference category. Multicollinearity was assessed based on the standard errors and model convergence. The results were reported as odds ratios (ORs) with 95% confidence intervals (CIs) and *p*-values. The model’s overall significance was evaluated using the LR Chi-square Test, and Nagelkerke’s pseudo R^2^ was used to assess the explanatory power.

All analyses were two-tailed, with a significance threshold of *p* < 0.05.

## 3. Results

### 3.1. Sample Characteristics

The final analytical sample consisted of 2090 respondents, of whom, 1645 (78.7%) had complete health literacy data based on the HLS-EU-Q16 instrument and 445 (21.3%) had incomplete scores and were excluded from the clustering analysis (Table 1). The mean age of the total sample was 52.7 years (SD 17.26), and 54.6% were female, with no significant difference in gender distribution between the complete and incomplete groups (*p* = 0.361). The age distribution differed significantly (*p* = 0.041), with younger respondents (18–34 years) more likely to have completed the HLS-EU-Q16, while older participants (≥55 years) were more frequently represented in the incomplete group. Similarly, the mother tongue was significantly associated with completion status (*p* < 0.001), with German speakers being more likely to have missing HLS-EU-Q16 data, whereas Italian speakers were more likely to have complete data.

There was no significant difference in citizenship status (*p* = 0.083) or urban versus rural residence (*p* = 0.324) between the groups. However, educational attainment was strongly associated with completion (*p* < 0.001), as those with middle or vocational school education were overrepresented among the respondents with incomplete HLS-EU-Q16 data.

The respondents living alone were more likely to have incomplete health literacy data (*p* = 0.003), as were those who were not employed in the healthcare sector (*p* < 0.001). Trust in general practitioners (GPs) also differed significantly (*p* < 0.001), with respondents who reported lower trust more often falling into the incomplete group.

No significant differences were observed in the presence of chronic diseases (*p* = 0.603). However, self-rated health status differed slightly, with those in the incomplete group reporting a slightly broader variability (interquartile range IQR: 25 vs. 20; *p* < 0.001). Although patient activation (PAM-10) scores showed only modest differences, they were statistically significant (*p* < 0.001), whereas sleep quality (B-PSQI) scores did not differ between the groups (*p* = 0.732).

### 3.2. Clustering Procedure, Model Selection, Profiles, and Between-Cluster Comparisons

Model-based clustering using Gaussian mixture models was performed on four z-standardized indicators: health literacy, patient activation, sleep quality, and total healthcare utilization. Solutions with two to six clusters were compared using BIC and silhouette scores. Three distinct subgroups were identified among the participants with valid health literacy scores (*n* = 1645). The three-cluster solution showed favorable model characteristics, including a silhouette coefficient of 0.130 and moderate entropy (0.714), indicating acceptable internal cohesion and separation.

Cluster 1 (n = 1197, 72.8%) represented the largest group and was characterized by moderate patient activation, slightly above-average health literacy (z = +0.125), average sleep quality, and below-average primary and urgent care service utilization in GP offices or ERs (z = −0.369). This cluster likely represents a general population profile with stable health behaviors and self-management.Cluster 2 (*n* = 424, 25.8%) displayed the lowest health literacy scores (z = −0.247), slightly reduced activation, average sleep, and increased utilization of GP and ED services (z = +0.671). This group may reflect a vulnerable subgroup at risk of overutilization due to limited comprehension and self-management capacities.Cluster 3 (*n* = 24, 1.4%) was a small but distinct group marked by extremely high patient activation (z = +1.91), slightly elevated health literacy, and the highest utilization of healthcare services.

Kruskal–Wallis tests confirmed significant between-cluster differences across all clustering variables (*p* < 0.001), supporting the empirical distinctiveness of the three subgroups. Post hoc analyses indicated that health literacy significantly differed between Clusters 1 and 2, as well as between Clusters 2 and 3, highlighting its discriminative role.

Sociodemographic, behavioral, and health-related characteristics were compared across the three behavioral clusters (Table 2). For ease of interpretation and to reflect behavioral characteristics, the three clusters were labeled as ‘Balanced Self-Regulators’, ‘Struggling Navigators’, and ‘Hyper-Engaged Users’, respectively. Statistically significant differences were observed in most of the characteristics examined.

Health literacy, patient activation, and sleep quality were included as input variables in the clustering algorithm and thus inherently differentiated the identified clusters. Marked differences were observed across the clusters in terms of patient activation levels, health literacy, and sleep quality.

‘Balanced Self-Regulators’ were most likely to exhibit higher patient activation (PAM-10), with over one-fifth of the group reaching the highest activation level, characterized by maintenance of health behaviors and continued engagement. In contrast, the ‘Struggling Navigators’ were overrepresented in the lowest activation tiers, particularly in the category of becoming aware but still struggling. The ’Hyper-Engaged Users’, while a small group, showed a more evenly distributed activation profile, with a notably higher share in the top activation tier compared to the ‘Struggling Navigators’.

Health literacy (HLS-EU-Q16) followed a similar pattern. The majority of ‘Balanced Self-Regulators’ were categorized as having sufficient health literacy, while the ‘Struggling Navigators’ had the highest proportion of individuals with inadequate health literacy and the lowest share with sufficient scores. The ‘Hyper-Engaged Users’ showed a mixed profile, with a majority in the sufficient category and a smaller share in the inadequate range.

In terms of sleep quality (B-PSQI), a substantially higher proportion of ‘Struggling Navigators’ reported poor sleep compared to the ‘Balanced Self-Regulators’, who predominantly fell within the good sleep category. The ‘Hyper-Engaged Users’ again showed a more even split between poor and good sleep quality, indicating variability in sleep patterns despite high engagement levels.

The distribution of ‘Struggling Navigators’ across urban and rural areas largely reflected the underlying population distribution; no meaningful overrepresentation was observed when adjusted for base rates. Gender and citizenship status did not differ significantly across the clusters. In contrast, the ‘Struggling Navigators’ were more likely to report lower levels of formal education.

The results for external health-related variables revealed that the ‘Struggling Navigators’ and ‘Hyper-Engaged Users’ were more likely to report chronic disease and had reduced health status perception and poor sleep quality compared to ‘Balanced Self-Regulators’. While trust in general practitioners was generally high, it did not differ significantly across the groups.

The ‘Hyper-Engaged Users’ represented a small but behaviorally distinct cluster, characterized by very high patient activation, a greater proportion of urban residence and Italian-speaking participants, and a co-occurrence of elevated healthcare utilization with less favorable self-rated health.

### 3.3. Primary and Urgent Care Services Utilization Stratified by Behavioral Determinants

#### 3.3.1. Sleep Quality

The primary and urgent care services use differed significantly according to sleep quality (B-PSQI). The total number of primary and urgent care services contacts was higher in the poor sleep group (median, 3.0) than in the good sleep group (median, 2.0; *p* < 0.001). The effect sizes were small (ε^2^ = 0.008–0.012), yet the consistent pattern suggests that poor sleep is associated with an increased use of both general practice and emergency services.

#### 3.3.2. Health Literacy and Patient Activation

Primary and urgent care services utilization was analyzed in relation to both health literacy (HLS-EU-Q16) and patient activation (PAM-10) levels. Across all outcomes, GP visits, ER visits, and total GP + ER contacts, no statistically significant differences were found when stratifying by either health literacy category or patient activation level.

The mean total primary and urgent care services contact (GP plus ER) was slightly higher among individuals with sufficient health literacy (mean 4.25; SD 12.48) than among those with problematic (mean 3.66; SD 9.14) or inadequate health literacy (mean 3.92; SD 7.32), but these differences did not reach statistical significance (ANOVA, *p* = 0.806; Kruskal–Wallis, *p* = 0.419). Similarly, the mean contact by patient activation level ranged from 3.13 to 4.06, with no significant trends or group differences (ANOVA, *p* = 0.834; Kruskal–Wallis, *p* = 0.132). The effect sizes were consistently negligible (ε^2^ < 0.01), and no significant post hoc differences were observed.

### 3.4. Multinomial Logistic Regression of Sociodemographic and Health-Related Predictors of Cluster Membership

A multinomial logistic regression was conducted to identify sociodemographic and health-related predictors of behavioral cluster membership using the ‘Balanced Self-Regulators’ (Cluster 1) as the reference group. The model was statistically significant (χ^2^ = 278.46, df = 28, *p* < 0.001), with a Nagelkerke R^2^ value of 0.215 (Table 3).

The model included the following predictors: age (continuous), gender, education level, residence (urban vs. rural), citizenship, mother tongue (German, Italian, other), living alone (yes/no), employment in the healthcare or social sector (yes/no), presence of chronic disease (yes/no), and self-rated health status (dichotomized). Collinearity diagnostics (Condition Index and Variance Proportions) indicated that the multicollinearity among predictors was acceptable, with no critical violations detected.

LR tests were used to evaluate the overall contribution of each predictor to the multinomial logistic regression model. The strongest individual effects were observed for health status perception (χ^2^ = 60.23, df = 2, *p* < 0.001), presence of chronic disease (χ^2^ = 41.72, df = 2, *p* < 0.001), and mother tongue (χ^2^ = 27.28, df = 4, *p* < 0.001), all of which significantly improved the model fit when included. Gender (χ^2^ = 11.57, df = 2, *p* = 0.003) and employment in the healthcare or social sectors (χ^2^ = 10.06, df = 4, *p* = 0.039) also contributed significantly. In contrast, age was borderline significant (χ^2^ = 5.61, df = 2, *p* = 0.060), whereas variables such as education, urban vs. rural residence, citizenship, and living alone did not significantly improve the model and were retained or excluded based on stepwise criteria.

The ORs represent the relative likelihood of belonging to Cluster 2 (‘Struggling Navigators’) or Cluster 3 (‘Hyper-Engaged Users’) compared to Cluster 1 (‘Balanced Self-Regulators’) for each predictor change. An OR of 1.60 for Italian-speaking respondents indicates they had 60% higher odds of being classified as ‘Struggling Navigators’ compared to German speakers, when the other factors are constant. The respondents with a chronic disease were 58% less likely (OR = 0.425) to belong to Cluster 2 than Cluster 1, reflecting increased engagement among those managing chronic conditions. Cluster 3 ‘Hyper-Engaged Users’ were more likely to be older (OR = 1.010 per year) and report better health (OR = 0.661 for reduced health status), suggesting an actively involved subgroup with higher perceived well-being.

Thus, compared to ‘Balanced Self-Regulators’, ‘Hyper-Engaged Users’ were slightly older and less likely to live alone. They also reported a better overall health status. ‘Struggling Navigators’, in contrast, were more likely to be Italian speakers or to speak a language other than German and were less likely to live alone. The presence of chronic diseases was associated with a lower likelihood of belonging to this group. The respondents working in the healthcare sector had significantly lower odds of being in the ‘Struggling Navigators’ group compared to the ‘‘Balanced Self-Regulators’’. Education, gender, and urban versus rural residence were not significantly associated with cluster membership in this model.

## 4. Discussion

This study applied a model-based clustering approach to identify the latent behavioral profiles of primary healthcare users in a culturally diverse region of northern Italy. Using health literacy, patient activation, sleep quality, and total primary and urgent care service contact as input variables, a three-cluster solution was identified that provided a meaningful segmentation of the adult population.

The largest group, ‘Balanced Self-Regulators’ (72.8%), was characterized by moderate patient activation, sufficient health literacy, good sleep quality, and below-average service use. In contrast, ‘Struggling Navigators’ (25.8%) exhibited lower activation, higher prevalence of inadequate health literacy, poor sleep quality, and increased primary and urgent care service utilization, particularly emergency rooms. The smallest group, ‘Hyper-Engaged Users’ (1.4%), displayed high activation, elevated service use, and mixed indicators of health status and sleep quality, suggesting possible over engagement or complex care needs.

The descriptive and inferential comparisons confirmed that sleep quality was significantly associated with higher health care use. While health literacy and patient activation were key dimensions defining the clusters, they were not independently associated with primary and urgent care service contact when analyzed separately. Multinomial regression showed that language, presence of chronic diseases, and perceived health status were significant predictors of cluster membership. These findings suggest that a latent profile approach can reveal important behavioral and cognitive dimensions of healthcare engagement that are not apparent in traditional variable-centered analyses.

### 4.1. Interpretation and Comparison with Existing Literature

This study identified three distinct latent profiles of primary and urgent care service users based on health literacy, patient activation, sleep quality, and service utilization. These profiles reflect the known behavioral and psychosocial determinants of healthcare engagement.

Sleep quality was used as a behavioral indicator due to its well-documented associations with both mental health and chronic physical conditions, as well as its predictive value for primary and emergency care utilization across diagnostic groups. Consistent with previous research, poor sleep quality was significantly correlated with increased utilization of primary and emergency care services, particularly visits to GPs and ERs. This finding corroborates with previous studies that have demonstrated an association between sleep disturbances, especially insomnia, and heightened use of primary and emergency care, even when controlling for comorbidities ([24]; [31]; [37]; [41]). A previous scoping review identified poor sleep as an independent risk factor for elevated healthcare utilization and costs ([3]). In our dataset, individuals with poor sleep exhibited a higher frequency of both GP and ER contact, highlighting the importance of sleep quality as a modifiable determinant of service demand in primary and urgent care services.

In the context of health literacy, our findings corroborate that inadequate or problematic health literacy was a defining characteristic of the high-utilization group identified as ‘Struggling Navigators’. This aligns with research conducted in Sweden, Denmark, Switzerland, Japan, and the United States, which indicates that limited health literacy is associated with an increased frequency of physician visits, particularly in primary and urgent care services, and greater challenges in navigating health systems ([6]; [14]; [15]; [33]; [38]; [39]). However, in the stratified analysis of the South Tyrol sample, health literacy alone did not exhibit a significant association with the number of primary and urgent care services visits, suggesting that its impact on healthcare utilization is most pronounced when considered alongside other behavioral dimensions, such as patient activation and sleep quality, rather than as an isolated predictor.

Patient activation emerged as a significant dimension for behavioral clustering, with the ‘Hyper-Engaged Users’ exhibiting the highest activation scores. This finding aligns with those of previous research ([8]; [21]), indicating that elevated patient activation is associated with enhanced self-management, increased uptake of preventive care, and a reduction in emergency visits and hospitalizations. However, in our stratified analyses, the PAM scores did not significantly differentiate service utilization. This may be indicative of contextual variations in the healthcare system structure, population characteristics, or cultural norms that affect how activation translates into care-seeking behavior ([12]; [26]).

Our findings underscore the interaction between health literacy and patient activation, which, while related, are conceptually distinct constructs. Prior research indicated that health literacy may affect activation, or alternatively, activation may mediate the effects of low literacy on service utilization ([17]; [19]). Within our clusters, these constructs co-occurred, albeit with varying influences across different profiles, thereby supporting the notion that multidimensional behavioral segmentation may offer a more comprehensive model of healthcare engagement compared to single-variable approaches.

However, this relationship appears to be context-dependent. [10] ([10]) found no significant association between health literacy and patient activation in a population of frequent users of primary and urgent care services, suggesting that this link may not be universally present across all patient groups. Their sample showed high activation levels alongside predominantly low health literacy, which may reflect the specific characteristics of vulnerable populations with intensive care needs. Similarly, in our study, although health literacy and activation co-occurred in certain clusters, particularly among ‘Struggling Navigators’, the two constructs differed in their influence on service utilization when analyzed independently. This finding supports the notion that multidimensional behavioral segmentation may capture the complexity of healthcare engagement more effectively than linear analyses based on single variables.

In summary, our results corroborate and expand upon the existing literature concerning health literacy, patient activation, and sleep quality as behavioral determinants of primary healthcare utilization. Through the application of LPA, this study elucidated how these dimensions cluster within distinct population subgroups, providing an improved perspective for understanding variations in healthcare behavior.

### 4.2. Implications for Practice and Policy

The findings of this study have implications for public health strategies and primary and urgent care services organizations. By identifying distinct behavioral profiles of healthcare users based on modifiable cognitive and psychosocial dimensions, such as health literacy, patient activation, and sleep quality, this study offers a framework for targeted, efficient, and equitable interventions.

First, the identification of the Struggling Navigators cluster—characterized by inadequate health literacy, lower activation, poor sleep, and elevated service use—underscores the need for integrated interventions that go beyond disease-specific management. Programs that combine health literacy education ([32]; [38]) with skill-based activation training ([4]) may empower individuals to manage their health more effectively. At the same time, efforts to strengthen organizational health literacy, by making health services easier to access, understand, and navigate, could reduce barriers for those with limited individual competencies. Such multi-level strategies are particularly important for reducing unnecessary contacts with primary and emergency care services among vulnerable subgroups. Similarly, sleep-focused interventions, including brief cognitive-behavioral or self-management approaches, can alleviate avoidable utilization triggered by non-restorative sleep and associated somatic complaints ([35]; [41]).

Second, the latent profile approach demonstrated the feasibility of behavioral segmentation in population health management. By incorporating brief instruments, such as the HLS-EU-Q16, PAM-10, and B-PSQI, into routine assessments or digital triage tools, health systems could enable the early identification of high-need individuals who are at risk of care overutilization or underutilization due to behavioral barriers ([9]; [13]). This proactive identification can inform case management or preventive outreach strategies.

Importantly, these efforts must be culturally and linguistically adapted to regional contexts, such as South Tyrol, where two main language groups (German- and Italian-speaking) coexist within a shared health system. In the adjusted regression model, Italian language background was associated with higher odds of being classified as a ‘Struggling Navigator’ compared to German speakers. This effect was not evident in unadjusted descriptive proportions but became apparent after accounting for other sociodemographic and health-related variables. This finding emphasizes the need to design materials and interventions that are not only translated but also culturally resonant, ensuring accessibility and trust across groups ([42]).

While individual predictors, such as health literacy and patient activation, did not independently explain service use in stratified models, they played a key role in shaping behavioral clusters. This supports the notion that multidimensional interventions targeting co-occurring behavioral traits may be more effective than single-focus programs ([44]). Public health policies should prioritize behavioral screening and cluster-informed care pathways to optimize resource allocation and reduce strain in primary and urgent care services systems.

Health policymakers can embed behavioral segmentation into service planning and outreach. ‘Struggling Navigators’, who show low health literacy, activation, and poor sleep, could be identified via brief screening (e.g., with HLS-EU-Q16 or PAM-10) during GP visits or digital pre-registration. Tailored interventions could include simplified health materials, peer navigation support, and the integration of sleep self-management guidance into chronic disease programs ([30]). At a system level, improving organizational health literacy by simplifying service pathways and communication can reduce reliance on emergency care for non-urgent issues. Health promotion campaigns should be designed to reach specific linguistic and behavioral subgroups, ensuring that both German- and Italian-speaking populations engage with preventive and primary care services.

### 4.3. Strengths and Limitations

This study has key methodological strengths. It uses a representative population-based sample from South Tyrol, a culturally and linguistically diverse region, enhancing the relevance of the findings for multilingual healthcare settings. The use of validated instruments, including the HLS-EU-Q16 for health literacy, PAM-10 for patient activation, and B-PSQI for sleep quality, ensures precision and reliability. The application of multivariate model-based clustering using GMMs allowed for the theoretically grounded identification of latent behavioral profiles, providing insights beyond traditional regression methods.

However, this study had several limitations. The cross-sectional design restricts causal inferences between behavioral characteristics and healthcare utilization. Variables related to service use and sleep were self-reported, which may have introduced a recall bias.

A key methodological issue concerns missing health literacy data; 21.3% of participants had incomplete HLS-EU-Q16 responses and were excluded from the cluster analysis. As shown in Table 1, non-respondents were older, more likely to speak German, and had lower education levels. This introduces potential selection bias, as excluded individuals may represent a group with greater needs, limiting the sample’s representativeness and possibly resulting in an underestimation of the low health literacy prevalence. The ‘Struggling Navigators’ profile may reflect a more engaged subset of the true at-risk population.

The Gaussian Mixture Modeling technique assumes that continuous variables are normally distributed within each latent class. When z-standardized input variables were used, their distributions did not fully meet the normality assumptions. Despite this, the model fit indices supported the three-cluster solution, and the resulting profiles were conceptually coherent. However, deviation from the model assumptions affects the robustness of the clustering results.

Multicollinearity among predictors in the multinomial logistic regression model was assessed using collinearity diagnostics and Spearman’s correlation coefficients. No severe multicollinearity was detected. However, moderate associations were observed between age and other health-related variables, such as chronic disease presence and self-rated health status. These interrelationships likely explain why age did not independently predict cluster membership after adjustment, as its influence was partially mediated or absorbed by health status and chronic disease burden. Future studies could explore alternative modeling approaches (e.g., structural equation modeling) to better disentangle the direct and indirect effects of age on behavioral profiles.

Finally, the small size of Cluster 3 (‘Hyper-Engaged Users’), although theoretically meaningful, limits statistical comparison precision and increases the risk of overfitting. Replication in a larger sample is essential to confirm the stability and generalizability of this behavioral segment.

### 4.4. Future Research Directions

Future research should employ longitudinal methodologies to elucidate the causal relationships between behavioral determinants and healthcare utilization. To enhance measurement validity, integrating survey data with electronic health records (EHRs) would facilitate the validation of self-reported service use. Considering the cultural and linguistic distinctiveness of South Tyrol, replication in other diverse contexts is necessary to evaluate its generalizability. Furthermore, intervention studies addressing cluster-specific barriers, such as low health literacy, inadequate sleep, or low activation, should be developed and assessed to inform personalized needs-based strategies in primary and urgent care services.

## 5. Conclusions

This study delineated three distinct behavioral profiles among primary and urgent care services users, characterized by variations in health literacy, patient activation, sleep quality, and service utilization. The findings underscore the utility of latent profiling in revealing patterns of healthcare engagement that remain obscured by traditional variable-centered methodologies. Notably, poor sleep quality emerged as a significant factor correlated with increased service utilization, while low health literacy and patient activation were prominent characteristics of the high-need groups. These results emphasize the necessity of incorporating behavioral and psychosocial assessments into primary and urgent care services, particularly within culturally diverse populations, such as South Tyrol. Enhancing sleep quality and health literacy and activation through targeted interventions may contribute to reducing avoidable service utilization and fostering more equitable access. Behavioral segmentation offers a promising framework for tailoring public health strategies to specific needs of population subgroups. Behavioral profiling can support more equitable resource allocation by targeting modifiable traits—such as low activation or poor sleep—through low-threshold, tailored interventions embedded in primary care, health communication, and digital triage tools.

## Figures and Tables

**Figure 1 behavsci-15-00724-f001:**
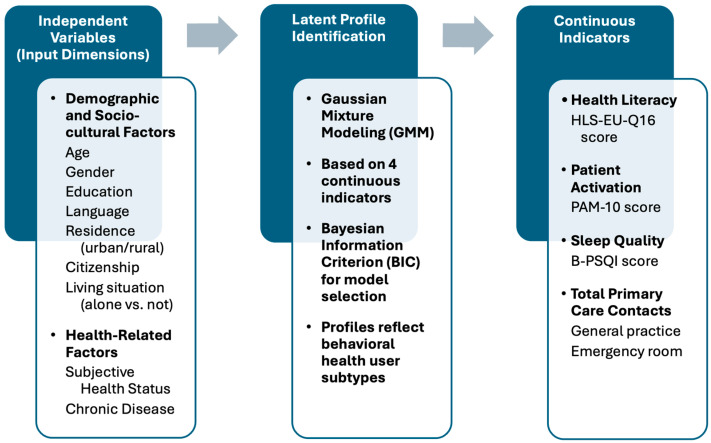
Conceptual framework of latent profile identification based on behavioral health determinants and service use in primary and urgent care services.

**Table 1 behavsci-15-00724-t001:** Sociodemographic and health-related characteristics of the sample and HLS-EU-Q16 completion status.

Variable	Total *n* = 2090	HLS-EU-Q16 Score Complete, *n* = 1645	HLS-EU-Q16 Score Incomplete, *n* = 445	*p*-Value ^1^
Gender, *n* (%)				0.361
male	937 (44.8%)	746 (45.4)	191 (42.9)	
female	1153 (55.2%)	899 (54.6)	254 (57.1)	
Age group (years), *n* (%)				0.041
18–34	383 (18.3)	313 (19.0)	70 (15.7)	
35–54	647 (31.0)	521 (31.7)	126 (28.3)	
55–99	1060 (50.7)	811 (49.3)	249 (56.0)	
Mother tongue, *n* (%)				<0.001
German	1398 (66.9)	1074 (65.3)	324 (72.8)	
Italian	499 (23.9)	420 (25.5)	79 (17.8)	
Ladin	82 (3.9)	71 (4.3)	10 (2.2)	
other, more than one	112 (5.3)	80 (4.8)	32 (7.2)	
Citizenship, *n* (%)				0.083
Italian	2011 (96.2)	1589 (96.6)	422 (94.8)	
other	79 (3.8)	56 (3.4)	23 (5.2)	
Residence, *n* (%) ^2^				0.324
urban	381 (18.2)	307 (18.7)	74 (16.6)	
rural	1709 (81.8)	1338 (81.3)	371 (83.4)	
Level of education, *n* (%)				<0.001 ^2^
middle school	492 (23.5)	339 (20.6)	153 (34.4)	
vocational school	674 (32.3)	520 (31.6)	154 (34.6)	
high school	530 (25.4)	432 (26.3)	98 (22.0)	
university	394 (18.8)	534 (21.5)	40 (9.0)	
Living alone, *n* (%)				0.003
yes	383 (18.3)	280 (17.0)	103 (23.2)	
no	1707 (81.7)	1386 (83.0)	342 (76.8)	
Healthcare worker, *n* (%)				<0.001
yes	217 (10.4)	194 (11.8)	23 (5.2)	
no	1872 (89.6)	1450 (88.2)	422 (94.8)	
Trust in GP, *n* (%)				<0.001
very much	893 (42.7)	727 (44.2)	166 (37.3)	
some	961 (46.0)	752 (45.7)	209 (47.0)	
a little	193 (9.2)	140 (8.5)	53 (11.9)	
not at all	43 (2.1)	26 (1.6)	17 (3.8)	
Chronic disease, *n* (%)				0.603
yes	795 (38.0)	621 (37.8)	174 (39.1)	
no	1295 (62.0)	1024 (62.2)	271 (61.9)	
Health status, median [IQR] ^3^	80.0 [20.0]	80.0 [20.0]	80.0 [25.0]	<0.001
HLS-EU-Q16 score, median [IQR] ^4^	n.d.	12.0 [5.0]	n.d.	—
PAM-10 score, median [IQR] ^5^	52.9 [12.6]	52.9 [12.6]	51.0 [9.3]	<0.001
B-PSQI score, median [IQR] ^6^	3.0 [4.0]	3.0 [4.0]	3.0 [4.0]	0.732

^1^ Chi-square tests for categorical variables and Kruskal–Wallis tests for continuous or ordinal variables. ^2^ Urban residence refers to respondents living in the regional capital, Bolzano/Bozen; rural residence includes all other areas in South Tyrol. ^3^ Health status was self-rated on a visual analogue scale from 0 (“worst imaginable health”) to 100 (“best imaginable health”). ^4^ HLS-EU-Q16: European Health Literacy Survey Questionnaire, 16-item short form; values range from 0 (lowest) to 16 (highest health literacy); n.d., not determined: HLS-EU-Q16 score was only calculated for respondents with sufficient item completion (≥13 valid responses). ^5^ PAM-10: Patient Activation Measure; range: 0–100. ^6^ B-PSQI: Brief Pittsburgh Sleep Quality Index; range: 0–21, with higher scores indicating worse sleep quality.

**Table 2 behavsci-15-00724-t002:** Comparison of sociodemographic, behavioral, and health-related characteristics across behavioral clusters.

Variable	1—Balanced Self-Regulators (*n* = 1197)	2—Struggling Navigators (*n* = 424)	3—Hyper-Engaged Users (*n* = 24)	*p*-Value ^6^
Gender, *n* (%)				0.059
male	564 (47.1)	173 (40.8)	9 (37.5)	
female	633 (52.9)	251 (59.2)	15 (62.5)	
Age group (years), *n* (%)				
18–34	249 (20.8%)	61 (10.4%)	3 (9.4%)	<0.001
35–54	416 (34.8%)	100 (17.1%)	5 (15.6%)	
55–99	532 (44.4%)	424 (72.5%)	24 (75.0%)	
Mother tongue, *n* (%)				<0.001
German	808 (67.5%)	263 (62.0%)	3 (12.5%)	
Italian	282 (23.6%)	119 (28.1%)	19 (79.2%)	
Ladin	56 (4.7%)	14 (3.3%)	1 (4.2%)	
other	31 (2.6%)	17 (4.0%)	0 (0.0%)	
more than one language	20 (1.7%)	11 (2.6%)	1 (4.2%)	
Citizenship, *n* (%)				0.925
Italian	1156 (96.6%)	410 (96.7%)	23 (95.8%)	
other	41 (3.4%)	14 (3.3%)	1 (4.2%)	
Residence, *n* (%) ^1^				0.009
urban	213 (17.8%)	84 (19.8%)	10 (41.7%)	
rural	984 (82.2%)	340 (80.2%)	14 (58.3%)	
Level of education, *n* (%)				<0.001
middle school	205 (17.1%)	128 (30.2%)	6 (25.0%)	
vocational school	378 (31.6%)	138 (32.5%)	4 (16.7%)	
high school	325 (27.2%)	100 (23.6%)	7 (29.2%)	
university	289 (24.1%)	58 (13.7%)	7 (29.2%)	
Living alone, *n* (%)				0.934
no	999 (83.5%)	346 (81.6%)	20 (83.3%)	
yes	198 (16.5%)	78 (18.4%)	4 (16.7%)	
Healthcare worker, *n* (%)				0.003
yes	161 (13.4%)	31 (7.3%)	2 (8.3%)	
no	1036 (86.6%)	392 (92.7%)	22 (91.7%)	
Trust in GP, *n* (%)				0.575
very much	525 (43.9%)	190 (44.8%)	12 (50.0%)	
some	552 (46.1%)	190 (44.8%)	10 (41.7%)	
a little	105 (8.8%)	33 (7.8%)	2 (8.3%)	
not at all	15 (1.3%)	11 (2.6%)	0 (0.0%)	
Chronic disease, *n* (%)				<0.001
no	846 (70.7%)	168 (39.6%)	10 (41.7%)	
yes	351 (29.3%)	256 (60.4%)	14 (58.3%)	
Health status perception, *n* (%) ^2^				<0.001
positive	861 (71.9%)	169 (39.9%)	11 (45.8%)	
reduced	336 (28.1%)	255 (60.1%)	13 (54.2%)	
Patient activation (PAM-10), *n* (%) ^3^				<0.001
disengaged and overwhelmed	160 (13.4%)	90 (21.2%)	4 (16.7%)	
becoming aware but still struggling	474 (39.6%)	192 (45.3%)	8 (33.3%)	
taking action	305 (25.5%)	86 (20.3%)	6 (25.0%)	
maintaining behaviors and pushing further	258 (21.6%)	56 (13.2%)	6 (25.0%)	
Health literacy (HLS-EU-Q16), *n* (%) ^4^				<0.001
inadequate	111 (9.3%)	144 (34.0%)	3 (12.5%)	
problematic	418 (34.9%)	143 (33.7%)	6 (25.0%)	
sufficient	668 (55.8%)	137 (32.3%)	15 (62.5%)	
Sleep quality (B-PSQI), *n* (%) ^5^				<0.001
poor sleep (score > 5)	211 (17.6%)	229 (54.0%)	12 (50.0%)	
good sleep (score ≤ 5)	986 (82.4%)	195 (46.0%)	12 (50.0%)	

Values are presented as absolute numbers with percentages, unless otherwise indicated. Percentages refer to the proportions within each cluster. ^1^ Urban residence refers to participants residing in the regional capital (Bolzano/Bozen); rural areas include all other areas. ^2^ Health status was dichotomized using the sample median (80) on a scale ranging from 0 (worst imaginable health) to 100 (best imaginable health). ^3^ Patient activation was categorized into four levels based on PAM-10; score ranges: Level 1, 0.0–<47.0; Level 2, 47.0–<55.1; Level 3, 55.1–<67.0; Level 4, ≥67.0 ([22]). ^4^ Health literacy was categorized using HLS-EU-Q16 scores: 0–8 = inadequate, 9–12 = problematic, and 13–16 = sufficient. ^5^ Sleep quality was assessed using the Brief Pittsburgh Sleep Quality Index (B-PSQI); scores >5 indicate poor sleep quality. ^6^
*p*-values were derived from the chi-square tests.

**Table 3 behavsci-15-00724-t003:** Multinomial logistic regression analysis of predictors of behavioral cluster membership.

Cluster ^1^	Predictor	B (Log Odds) ^2^	Odds Ratio ^3^	95% CI ^4^	*p*-Value
Hyper-Engaged Users	Age (years)	0.010	1.010	1.001–1.018	0.023
	Reduced health status	−0.414	0.661	0.515–0.849	0.001
	Living alone (yes)	−0.516	0.597	0.385–0.927	0.022
Struggling Navigators	Mother tongue: Italian	0.473	1.605	1.058–2.434	0.026
	Mother tongue: Other	0.399	1.491	1.022–2.175	0.038
	Chronic disease (yes)	−0.856	0.425	0.327–0.554	<0.001
	Living alone (yes)	−0.994	0.370	0.287–0.476	<0.001
	Reduced health status	−0.426	0.653	0.416–1.026	0.064
	Work in healthcare (yes)	–0.515	0.597	0.385–0.927	0.022

^1^ Cluster 1, ‘Balanced Self-Regulators’ (reference group); Cluster 2, ‘Hyper-Engaged Users’; Cluster 3, ‘Struggling Navigators’.^2^ B, unstandardized log-odds coefficient from the multinomial logistic regression model. ^3^ OR, odds ratio (Exp(B)); values > 1 indicate higher odds of cluster membership relative to the reference group. ^4^ CI, confidence interval.

## Data Availability

Data are available from the corresponding author upon request.

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
