# Peer review of "Primary and Emergency Care Use: The Roles of Health Literacy, Patient Activation, and Sleep Quality in a Latent Profile Analysis"

_behavsci, 2025, doi:10.3390/bs15060724_

Round 1

Reviewer 1 Report

Comments and Suggestions for Authors

Review: "Frontline Healthcare Services Utilization Patterns: The Roles of Health Literacy, Patient Activation, and Sleep Quality in a Latent Profile Analysis"

The paper studies healthcare users (HCu) in a region (South Tyrol). It discusses creating clusters of HCu using healthcare services, health literacy, sleep quality, and patient activation. It finds that 3 clusters are sufficient to model latent profiles across this population.

The paper contributes to the discussion in public health to develop interventions tailored for specific groups according to their behavioral segmentation. The paper relies on previous research published in Wiedermann et al. (2025).

The methodology is sound, and using a Gaussian Mixture Modelling and logistic regression to predict cluster membership is adequate.

If the paper is revised correctly, it would make an interesting contribution to the literature. Below, I recommend several improvements:

  1. Provide more justification and motivation for focusing only on "South Tyrol".
  2. There are several alternatives to the questionnaires employed in this research. Provide more justification for the choices made.
  3. Why focus on sleep quality and not directly on mental health? The explanation for the choice of sleep quality is related to its impact on mental health and increased service use. Focusing on mental health could provide additional insights.
  4. Sections 3.4 (and Table 3) must explain the multinomial logistic results more. Please describe in detail how we can interpret the ODDS ratio results.
  5. It is challenging to understand the implications of the results for improving and designing public health policies. Improve the discussion. What should health policymakers do?

Reviewer 2 Report

Comments and Suggestions for Authors

The article reports on an interesting alternative approach to establishing the relationship between health literacy, patient activation, sleep quality and frequency of contact with healthcare services, not by examining the impact of the first three factors independently but in a combined ‘latent profile analysis’. The starting point is a self-reported survey obtained earlier from healthcare users in the area of  South Tyrol.

The introduction is well-presented, defining the various factors and explaining their potential importance on the basis of previous research (adding appropriate references).

Next, the research set-up is explained and the results are presented. Three clusters of healthcare users emerge from those results: the ‘balanced self-regulators’,  the ‘struggling navigators’ and the ‘hyper-engaged users’.

The statistical approach is explained in detail. As I indicated when accepting to review, I am insufficiently qualified to assess the finer points of the statistic analysis. However, I am worried about the significance of the figures in Table 2 and their interpretation.

Two examples may clarify my doubts:

Line 287 says: “The ‘Struggling Navigators’ were more likely to reside in rural areas”. This is possibly based on the figures in Table 2, which show that 84 respondents in this category hail from an urban area and 340 (80.2%) from a rural area. However, a fair comparison would need to take into account the total numbers of urban and rural respondents of the survey as shown in Table 1. It then appears that there are 84 ‘urban strugglers’ on a total of 381 (22%) and 340 ‘rural strugglers’ out of 1709 (19.9%), meaning the percentages are not that different and, if anything, show the reverse of what is claimed.

Similarly line 470 says: “Our findings showed that language background was a significant predictor of behavioural profile membership.” Table 2 shows that 263 (62%) of the ‘strugglers’ are German speakers. However, the correct ratio is 263 out of 1398 German speakers, which is only 18.8%, as compared against 144 ‘strugglers’ among 693 non-German speakers, yielding a very similar, and actually slightly higher, percentage of 20.8%.

If I am right with my interpretation, the results in the article will have to be reconsidered in this light. I have therefore recommended a so-called "major revision" but I think it is a revision that can quickly be implemented.

For the rest, this is a well-structured and well-written article. Some remaining remarks:

- (It is of course a pity that 445 surveys were discarded because they had incomplete responses;  the authors do admit that this is a sad limitation; it is forgivable).

- The consecutive paragraphs starting with lines 414 and  422 respectively begin with the same sentence and continue with a very similar one. Presumably this is an uncorrected part based on two competing draft versions. It needs looking into.

- The frequently used term utilization is sometimes spelt with z, sometimes with s. Proofreading is necessary to guarantee consistency in this word and possibly also in other words with -ize/-ise.

- some article titles display multiple capitals (Adding A Measure Of Patient Self-Management Capability To etc.), whilst others don’t. Here too, consistency is preferable (and will, I guess, depend on the editor’s style rules).

Round 2

Reviewer 1 Report

Comments and Suggestions for Authors

All the suggestions have been incorporated in the final version. 

Reviewer 2 Report

Comments and Suggestions for Authors

Glad to see that the text has been adjusted.